# Nerves and Pancreatic Cancer: New Insights into A Dangerous Relationship

**DOI:** 10.3390/cancers11070893

**Published:** 2019-06-26

**Authors:** Giulia Gasparini, Marta Pellegatta, Stefano Crippa, Marco Schiavo Lena, Giulio Belfiori, Claudio Doglioni, Carla Taveggia, Massimo Falconi

**Affiliations:** 1Division of Experimental Oncology, Pancreas Translational & Clinical Research Center, Istituto di Ricovero e Cura a Carattere Scientifico (IRCCS) San Raffaele Scientific Institute, 20132 Milan, Italy; 2Axo–Glial Interaction Unit, INSPE, Division of Neuroscience, Istituto di Ricovero e Cura a Carattere Scientifico (IRCCS) San Raffaele Scientific Institute, 20132 Milan, Italy; 3 Department of Pathology, Vita Salute San Raffaele University, 20132 Milan, Italy; 4Pathology Unit, Division of Experimental Oncology, Istituto di Ricovero e Cura a Carattere Scientifico (IRCCS) San Raffaele Scientific Institute, 20132 Milan, Italy

**Keywords:** pancreatic cancer, perineural invasion, neurotrophins, molecular signaling

## Abstract

Perineural invasion (PNI) is defined as the presence of neoplastic cells along nerves and/or within the different layers of nervous fibers: epineural, perineural and endoneural spaces. In pancreatic cancer—particularly in pancreatic ductal adenocarcinoma (PDAC)—PNI has a prevalence between 70 and 100%, surpassing any other solid tumor. PNI has been detected in the early stages of pancreatic cancer and has been associated with pain, increased tumor recurrence and diminished overall survival. Such an early, invasive and recurrent phenomenon is probably crucial for tumor growth and metastasis. PNI is a still not a uniformly characterized event; usually it is described only dichotomously (“present” or “absent”). Recently, a more detailed scoring system for PNI has been proposed, though not specific for pancreatic cancer. Previous studies have implicated several molecules and pathways in PNI, among which are secreted neurotrophins, chemokines and inflammatory cells. However, the mechanisms underlying PNI are poorly understood and several aspects are actively being investigated. In this review, we will discuss the main molecules and signaling pathways implicated in PNI and their roles in the PDAC.

## 1. Introduction

Pancreatic ductal adenocarcinoma (PDAC), the most common cancer of the exocrine pancreas, is known for its extreme aggressiveness with a 5–year survival rate below 10% [1] and it is estimated to become the second leading cause of death by 2030 [2,3].

One of the pathological hallmarks of PDAC is perineural invasion (PNI). Described in pancreatic cancer for the first time in 1944 [4] as a passive process of diffusion along planes of least resistance in the connective nerve sheath [5], it has been since extensively studied in several cancers [6].

Today, PNI is considered a more elaborate process and it is defined as the presence of cancer cells along nerves and/or within the epineural, perineural and endoneural spaces of the neuronal sheath, including cases in which the cells circumscribe at least 33% of the nerves [6,7].

Although present in several solid tumors, PNI has its highest prevalence in PDAC [8], with a range varying between 70% and 98%. Accordingly, it has been suggested that PNI could be detected in 100% of cases if enough pathological sections are examined [9,10,11,12,13,14]. Moreover, PNI is detected in nearly 75% of the early stages of PDAC and in microscopic PDAC, suggesting that it could represent an early event in cancer progression [6,7,12,15,16].

PNI can therefore be considered a characteristic and pervasive feature of PDAC [17] and such a widespread phenomenon probably has a role in supporting tumor growth. However, despite several new discoveries, a comprehensive and functional understanding of PNI is missing. Understanding the mechanisms by which PNI spreads is critical for developing targeted strategies directed not only towards cancer cells but also to the nerve microenvironment and possibly supporting cells, which likely have a crucial role in promoting tumor growth.

In this review, we will highlight clinical–pathological features of neural invasion as well as recent developments in its molecular comprehension.

## 2. Clinical Impact of Perineural Invasion

Although PNI is almost omnipresent in PDAC patients, a clear consensus in terms of the profound impact of perineural invasion on overall survival, disease free survival, early tumor recurrence is still missing. Numerous studies have reported a correlation between PNI and tumor recurrence and/or patients’ survival [8,9,12,18,19,20,21,22,23,24]; however, few studies failed to show PNI as an independent predictor of tumor recurrence (overall and early) and of patient survival [25,26]. These differences are most likely due not only to the retrospective design of the majority of these studies but also to the high variance in histological evaluation of PNI. Indeed, PNI is commonly classified as “present” or “absent” without further sub–classifications. However, at histological level, PNI is highly heterogeneous and it can be classified as intra–pancreatic intratumoral, intra–pancreatic extra tumoral or extra–pancreatic retroperitoneal [9].

A highly debated topic is whether neoplastic spread along nerves might constitute a route for lymph node metastasis [27,28,29]; this aspect could be particularly relevant for large caliber axons that are often in contact with lymph nodes. Of note, in the absence of both perineural and lymphovascular invasion, the 5–year survival rate for pancreatic adenocarcinoma patients rises to 71% [21]. Indeed, neoplastic spread creates a route outward the pancreas that may favor both local and systemic recurrence.

Another common symptom in PDAC patients is intractable pain. Pain is a negative prognostic factor for survival [30,31] and was firstly associated with PNI by Zhun Z. [32]. Cancer invasion damages the neural sheath and correlates with increased neural density and remodeling, resulting in both neuropathic and inflammatory pain [18,33,34]. Not surprisingly, many molecules that contribute to PNI are also responsible for pain onset. Nerve growth factor (NGF) released by cancer can sensitize sensory nerves and act on TRPV1 (Transient Receptor Potential Vanilloid 1), which in turn correlates with severe pain in patients [30,35,36]. Similarly, Glial cell line–Derived Neurotropic Factor (GDNF) and artemin (ARTN), released by cancer, increase TRPV1 expression and thus pain sensibility [37]. Activated TRPV1 then orchestrates the release of substance P and Calcitonin Gene–Related Peptide (CGRP) by nerves: both molecules contribute to neuropathic pain and their expression is increased in PNI [38].

Increased neural invasion also correlates with higher glucose levels in patients [39,40]. Hyperglycemia, present in more than 80% of patients, [41,42], supports tumor growth, spread and upregulation of NGF, which in turns enhances neural invasion [43,44] and axonogenesis directed at the neoplastic front [39]. Furthermore, in hyperglycemic conditions, nerves degenerate and Schwann cells migration and proliferation are impaired [45,46], making them more vulnerable to neoplastic invasion.

Notably, the anatomic location of the pancreas, surrounded by several neural plexuses [47,48,49] and the strong neurotropism of PDAC cells (Figure 1), can only partially explain such a high incidence of PNI. Indeed, not all pancreatic cancers have the same degree of neural invasion. PNI is less frequent in Intraductal Papillary Mucinous Neoplasm (IPMNs) with invasive carcinoma and neuroendocrine tumors (NETs)—though it has been recently demonstrated that PNI in NETs is a strong predictor of aggressiveness [50].

In this scenario, a recent study has proposed a new method to classify neural invasion in gastrointestinal malignancies with the aim of uniformly characterizing PNI; interestingly, in this study, among all tumors, PDAC emerged as the highest neuroaffin tumor [8].

## 3. Pancreatic Innervation

The Peripheral Nervous System (PNS) collects somatosensory information from the body and regulates physiological functions. Since tumors are not independent organs, they are innervated by peripheral nerves that are part of the tumor microenvironment [51]. Interestingly, nerves can adapt to the ongoing neoplastic processes and establish a bi–directional communication with the tumor, supporting their growth [52]. Further, peripheral nerves are an attracting conduit for neoplastic cells; indeed, cancer cells can be found in all the three different layers of nerves–endoneurium, perineurium, epineurium [17] (Figure 2)—not only superficially tracking along the nerve. The active role of PNS in supporting cancer is evidenced by in vivo preclinical studies. Experimental denervation of different organs in fact impairs tumor formation in animal models of prostate [53], gastric [54] and pancreatic [55] cancer.

The pancreas is a highly innervated organ: the coeliac plexus—the largest autonomic plexuses that lies between the pancreas and the superior mesenteric artery—innervates the head of the pancreas. The coeliac plexus includes right and left coeliac ganglia and the superior mesenteric ganglion. It gives rise to the main structure innervating the pancreas: the pancreaticus capitalis I plexus, derived from the coeliac plexus and the pancreaticus capitalis II plexus, derived from the superior mesenteric ganglion. The splenic plexus instead innervates the body and tail of the pancreas [11].

Both cadaveric studies and more recent imaging analyses with multi–detector row computer tomography have demonstrated a pattern of PNI based on the location of the tumor: tumors of the head expand towards the celiac plexus and ganglion along the pancreaticus capitalis I plexus. Uncinate process cancer instead expand towards the superior mesenteric plexus along inferior pancreaticoduodenal artery plexus, while pancreatic body and tail cancer expands towards the splenic and the celiac plexi [47,48,49,59,60].

## 4. Models to Study Perineural Invasion

Our understanding of the pathogenesis of PNI has been limited by a lack of effective models allowing a thorough analysis of the complex interactions between nerve, tumor cells and stroma (see BOX 1 for the mostly used models to study PNI).

### 4.1. In Vitro Models

In vitro models rely on co–cultures of pancreatic cancer cells and dorsal root ganglia (DRGs) sensory neurons [61]. This co–culture system allows the analyses of mutual interaction and attraction occurring between these cell types as well the investigation of the mechanisms controlling migration and tropism between cancer cells and nerves [62,63,64].

An ex vivo model with rat vagal nerves has also been developed [65]. This system allows the study of only invasive cells, as it selects neoplastic clones that, by passing through the nerve, reach the culture plate where they are specifically expanded. This is a valid method for investigating the differences between invasive and non–invasive pancreatic cancer cells.

More recently, pancreatic organoids have been developed and they represent an innovative model for studying PDAC. Literally “organ–like,” organoids are three–dimensional cell cultures growing inside an extracellular matrix that reproduces the tumoral architecture as well as cell–cell and cell–stroma interactions. Importantly, organoids can overcome many limitations found in monolayer cultures, such as tumor heterogeneity, tissue architecture and mechanical signals that in turn impact gene expression [66,67,68]. In addition, they could be co–cultured with other cell types, such as fibroblasts, immune cells and nerves, allowing the monitor of the interactions between different cell types in physiological conditions as well as in response to drugs [67,69,70,71,72,73].

### 4.2. In Vivo Models

Various mouse models that recapitulate PDAC are also available [74,75]. As highlighted by Demir and colleagues [34], these models should recapitulate neural sprouting, neural hypertrophy, pancreatic neuritis or neural invasion as it occurs in humans, to allow a comprehensive investigation of PNI, although a perfect neuroanatomic similarity has yet to be achieved.

Among the best–characterized models closely recapitulating all PDAC features observed in humans, the KPC mouse model (Pdx–1–Cre; LSL–Kras^G12D/+^ LSL–Trp53^R172H/+^) [74] was useful to determine the development of neuroplastic alterations during cancer progression [15]. These mutants develop cancer lesions strongly resembling human PDAC by 10 weeks of age. The carcinomas express CK19, a marker of disease present in almost 90% of patients [76] and mucin [75]. They also develop liver and lung metastasis similar to patients. Similarly, the KIC mouse model (Pdx1–Cre; LSL–Kras^G12D/+^; Ink4A^f/f^) [77] was used to study neural plasticity in response to PDAC and its microenvironment [78]. Since in this model IPMN tumors develop between 7 and 12 weeks of age [79], it is considered a valid model to study this form of cancer.

In addition to the above–described animal models, orthotopic models can facilitate the study of angiogenesis and metastasis [80] and, in the setting of PNI, the invasiveness of neoplastic cells along nerves [81]. In these models, cancer cells are injected into the pancreas (or the native organ) after being conjugated to fluorescent reporting proteins. This strategy allows the migratory behavior of cancer cells to be easily followed.

PNI can also be reproduced in several heterotopic models, where neoplastic cells are implanted in areas different from the original tumor site. These models can be useful for studying PNI, since injection of neoplastic cells into murine sciatic nerves [64,82,83,84,85] could recapitulate cancer cell migration and the invaded nerves can be collected and processed for further analyses. In addition, heterotopic models could facilitate the assessment of pain related to cancer invasion by performing behavioral tests. Further, they permit the imaging of cancer spreading by magnetic resonance studies [86]. On the contrary, they do not allow the study of early cancer lesions and the interactions between cancer and nerves at pre–neoplastic stages, since already established neoplastic cells are directly injected in the chosen area [34].

Box 1—PNI models

In vitro:
▪Co–cultures of DRGs and neoplastic cells: analyses of mutual interactions, tropism and migration.▪Ex–vivo model: selection of nerve invasive neoplastic cells and study of the differences with non–invasive tumor cells.▪Organoids: exact recapitulation of the original tumor structure. Can be co–cultured with other cell types to mimic cancer cells–nerve interactions.


In vivo:▪KPC mouse model (LSL–KrasG12D^/+^; LSL–Trp53R172H^/+^; Pdx–1–Cre+): it recapitulates the most common human PDAC phenotype.▪KIC mouse model (Pdx1–Cre; LSL–KrasG12D^/+^; Ink4Af/f): study of the accelerated development of PanINs into poorly differentiated PDAC.▪Orthotopic mouse models: study of the migratory behavior of neoplastic cells.▪Heterotopic mouse model: analyses of cancer cell migration and invasiveness.

## 5. Nerve—Cancer Microenvironment

Once cancer cells are inside the nerves, they create the tumor microenvironment. After invasion, tumor cells damage the nerves leading to inflammation and cycles of regenerative processes that are likely exploited by cancer to proliferate [20,65,83,87,88].

Interestingly, different cell types that are physiological components of the pancreas and are involved in tumorigenesis, support nerve–cancer interactions: fibroblasts are modified by cancer and secrete molecules that enhance neuroplasticity [78,89]; immune cells are reprogrammed by the tumor and promote neural invasion [83]; Schwann cells are attracted by cancer and facilitate neural tracking [90,91,92].

### 5.1. Fibroblasts

Cancer associated fibroblasts (CAFs), the main component of the tumor stroma, derive from bone marrow–derived mesenchymal stem cells, pancreatic stellate cells and stromal resident fibroblasts [93,94] and contribute to cancer invasiveness, angiogenesis and metastasis [93,95,96,97]. Fibroblasts are also the main cellular constituents of the perineurium [57] and damage to this structure alters blood nerve barrier permeability, thus increasing the risk of tumor cells’ invasion.

CAFs may also have a role in promoting neural invasion: they secrete Slit2, an axon guidance molecule, that through Robo1 and Robo2 receptors (via N–cadherin/β–catenin signaling) stimulates nerves remodeling and Schwann cells proliferation resulting in increased nerve density in in vitro PDAC models. Additionally, Slit2 expression and CAFs correlate with neural remodeling within human and mouse PDAC [78]. A very recent study proposed that macrophages also participate in the Slit–Robo signaling to control Schwann cells and axons migration after injury. Macrophages secret Slit3, which guides Schwann cells and fibroblasts in forming an appropriate and target–directed nerve bridge, suppressing aberrant migration [98]. Whether this mechanism also promotes nerve remodeling and Schwann cells proliferation in PNI, has yet to be investigated.

Of note, fibroblasts and macrophages present in the cancer microenvironment secrete the cytokine Leukemia Inhibitory Factor (LIF) [99,100], which can increase Schwann cells migration and neural plasticity leading to enhanced neurite outgrowth at least in vitro. Based on the results of this study, LIF has been proposed as a diagnostic marker for pancreatic cancer since it positively correlates with increased intra–tumoral neural density and, in combination with Ca19.9, could distinguish PDAC from other benign pancreatic diseases [89].

### 5.2. Pancreatic Stellate Cells

Stellate cells are myofibroblastic cells that are activated during tumorigenesis and support cancer progression and metastasis [101,102,103]. These cells could promote neurites’ outgrowth and therefore PNI, reinforce cancer cell migration toward nerves and facilitate extracellular matrix (ECM) degradation [7,30,104,105].

### 5.3. Immune Cells

In addition to altering the nerve structure, cancer cells can manipulate immune cells to favor tumor progression and metastasis [106,107,108,109,110]. This presumably also occurs in humans as an increased number of macrophages and PNI have been also reported in samples from PDAC patients [111]. Inflammatory changes have been observed in nerves already in the early stages of PDAC or even before in PanIN lesions, suggesting that nerves might modulate the immune system to support cancer progression [51,112], resulting in hypertrophy and hypersensitivity of pancreatic afferents and sensory fibers [113,114].

During tumorigenesis, cancer cells secrete Colony Stimulating Factor (CSF–1) to recruit macrophages; in turn, recruited macrophages release glial derived neurotrophic factor (GDNF) that activates RET (REarranged during Transfection) on cancer cells to promote cancer migration and nerve invasion [83].

In addition to the above–described molecules, the CCL2–CCR2 axis has also been implicated in PNI, as CCR2 deficient mice have reduced PNI and macrophage recruitment to the tumor site [83]. Indeed, it has been shown that CCL2 recruits monocytes and basophils to the inflammatory sites, which in turn express the receptor CCR2. Recent studies have demonstrated that in PNI, Schwann cells secrete CCL2 to recruit bloodstream monocytes, similar to what happens after a nerve injury insult [56,115,116]. These monocytes then differentiate into cathepsin B secreting macrophages that disrupt the perineurium, facilitating cancer invasion [117]. Of note, the therapeutic target of this axis is tolerable and can elicit an endogenous anti–tumorigenic immune response [118]. However, there are no indications as to whether this approach could also impact PNI.

### 5.4. Schwann Cells

Schwann cells are the glial cells of the PNS. They originate from neural crest cells and terminally differentiate either in myelinating or non–myelinating cells [115]. Myelinating Schwann cells associate with only one axon (α– and γ–motor axons and Aα–, Aβ– and Aδ–sensory axons) whereas non–myelinating Schwann cells associate with more than one C sensory and autonomic axons forming the structure called Remak Bundles [119]. Regardless of their final commitment, one of the most important roles of Schwann cells is to support neurons’ integrity. During cancer invasion, the nerve structure is altered and Schwann cells may have a role in promoting cancer cells’ adhesion to axons and thus their invasiveness (123). This phenomenon probably occurs because Schwann cells are plastic cells, as demonstrated by the analysis of the processes regulating nerve regeneration after injury [120].

A growing number of studies suggest that Schwann cells may have an active role in PNI in pancreatic cancer [90]: they have a specific affinity toward pre–neoplastic and neoplastic pancreatic cancer cells [91] and interestingly, after direct contact with cancer cells, Schwann cells actively promote cancer invasiveness and stimulate tumor protrusion and dispersion [92]. Despite the strong contribution of Schwann cells to neural invasion, their functional implications in invasiveness and in the development of pain are yet to be identified.

## 6. Molecules and Pathways Involved in PNI

Since Bockman and colleagues [17] hypothesized the first paracrine mechanisms of signaling between nerves and cancer cells to explain pancreatic cancer nerve–affinity, plenty of molecules have been implicated in PNI. What follows is a brief description of those that are involved in PNI, specifically in PDAC (see also Table 1).

### 6.1. Neurotrophic Factors and Neurotrophins’ Receptors

Neurotrophic factors and neurotrophins’ receptors enhance neuronal growth and survival [166,167] and play important roles in development, where they act as guidance cues for neurons. In the fully developed nervous system, they are involved in neuronal survival, synaptic plasticity and formation of long–lasting memories.

#### 6.1.1. Neurotrophins

The neurotrophins family of growth factors includes Nerve Growth Factor (NGF), Brain–Derived Neurotrophic Factor (BDNF), Neurotrophin–3 (NT–3) and Neurotrophin–4 (NT–4). These molecules bind to different receptors, a high affinity receptor (belonging to the TRK family) and the low affinity receptor p75NTR [168]. The binding of neurotrophins to their specific receptors results in autophosphorylation and the activation of several signaling pathways, among which Ras, phosphatidyl inositol–3 PI3–kinase/AKT, phospholipase C–γ1 and MAP kinase [169,170]. Whether in addition to mature neurotrophins, their precursors’ forms [171] also play a role in PDAC, has yet to be determined.

##### NGF and TrkA/p75NTR

Mature NGF is critical for the survival and maintenance of sympathetic and sensory neurons [172,173]. Of note, NGF is also produced in pancreatic cancer cells and its receptors TrKA and P75NTR are expressed on both pancreatic cancer cells and nerves, indicating that these molecules could play a role in PNI. Despite previous studies reported P75NTR as inversely correlated to PNI [44], more recently it has been recognized that P75NTR might act as a chemoattractant for cancer cells towards neural tissue to promote PNI [134]. Furthermore, NGF released by cancer cells promotes neuritic growth [7,62,135,136], reduces cancer cells’ apoptosis and increases proliferation leading to enhanced cancer aggressiveness [18,124,125,126,135]. Since Schwann cells also release NGF [90,127] it has been suggested that this growth factor might also be involved in enhancing glial cells migration in PNI [128].

As a further confirmation of the role of these molecules in pancreatic cancer, high levels of NGF and TrKA correlate with increased frequency and severity of PNI as well as reduced survival and increased pain in patients [7,44,124,129,130,131,132]. These events are also frequently associated with lymph node metastases [27]. Of note, NGF depletion by anti–NGF antibodies [112] or gene silencing with gold nanoclusters siRNA [133] reduces progression, metastasis and pain in several pre–clinical models of pancreatic tumors.

##### BDNF and TrkB

BDNF supports the survival of existing neurons and promotes the growth of new neurons and synapses [174,175]. BDNF and its high affinity receptors are overexpressed in metastatic human PDAC cells [137] but higher BDNF levels do not correlate with increased PNI [138]. Although in vitro studies have shown that BDNF overexpression is linked to increased proliferation and invasion of neoplastic cells and that inhibition of this axis reduced pancreatic cancer cells growth [124], in vivo analyses did not confirm it [139].

##### NT–3 and TrkC

NT–3 supports the growth and differentiation of already existing, as well as new, neurons [176]. NT–3 is overexpressed in human PDAC specimens [124] and both NT–3 and its receptor TrkC are highly expressed in intratumoral PDAC nerves [141]. Importantly, blocking NT–3 inhibited the growth of PDAC in a murine xenograft model [125], suggesting that this growth factor may have a role in invasiveness and pain generation [140].

#### 6.1.2. GDNF family

The importance of GDNF–RET signaling in perineural invasion in PDAC is well documented [31,64,129,142,177,178]. This family of growth factors consists of four members: glial cell line–derived neurotropic factor (GDNF), neurturin (NRTN), artemin (ARTN) and persephin (PSPN). They all bind to their cognate GDNF receptor α (GFRα–GFRα1, GFRα2, GFRα3, GFRα4 respectively) forming a complex that signals through the tyrosine kinase receptor RET. These complexes play a role in neural growth, differentiation, survival and nerve repair [179].

##### GDNF and GFRα1

GDNF is secreted by nerves and its activity can enhance both cancer cells aggressiveness and neural invasiveness. The role of this complex in PDAC tumors is underlined by the increased expression of both GFRα1 and RET receptors in patients [64,180,181].

It has been suggested that in PDAC, soluble GFRα1, which is released by nerves especially after injury [182], could facilitate the binding between neural GDNF and RET on pancreatic cancer, enhancing PNI [142,143,144]. In support of the critical role of this pathway in neural invasiveness, reducing GDNF expression or blocking RET activity reduces PNI both in vitro and in vivo [64].

Interestingly, perineural macrophages also express GDNF and tumor associated macrophages (TAMs) express more GDNF than resting macrophages [83]. Of note, GDNF secreted by M2 polarized recruited–perineurial macrophages, activates RET on cancer cells promoting PNI [52,83].

##### Neurturin and GFRα2

NRTN and its receptor GFRα2 are important in the normal development of pancreatic parasympathetic innervation [183] and their increased expression correlates with PDAC invasiveness and neural plasticity both in pancreatitis and PDAC [128,145].

##### Artemin and GFRα3

ARTN and its receptor GFRα3 are highly expressed in PDAC [146,147,184] and their level correlates with PNI severity. They are also upregulated in chronic pancreatitis after neural damage [146]. Recent studies have hypothesized that Artemin, initiates the migration of pancreatic cancer cells and promotes PNI and metastasis via NF–κB/CXCR4 signaling [148].

### 6.2. Chemokines

In addition to the CCL2/CCR2 axis we discussed above, several chemokines and their receptors are expressed in PDAC and implicated in cancer invasiveness and progression [185,186,187]. We will present those mainly involved in PNI.

#### CXCR1 and CX3CL1 (Fractalkin)

Both neurons and endothelial cells express CX3CL1 [188,189,190], the unique ligand for the receptor CX3CR1, which in the central nervous system (CNS) is implicated in neurons–microglia cross talk [191,192]. Interestingly, CX3CR1 is highly expressed in PDAC; moreover, it has been shown that these overexpressing cells migrate and adhere to nerves in response to both soluble and membrane–bound CX3CL1 [20]. In vivo in mice, CX3CR1–positive tumor cells infiltrate peripheral nerves while in PDAC patients, high receptor expression is associated with a prominent neural tropism and local recurrence [20,122].

Notably, both CX3CL1 and CX3CR1 are expressed in PanIN, indicating that these molecules might be involved in early cancer chemotaxis [123] to promote nerve invasion [121].

### 6.3. Axonal Guidance Molecules

During nerve–cancer interaction, chemo–attraction as well as repulsion can occur. In agreement, several studies have analyzed the role of some of these molecules in neural invasion in PDAC.

#### SLIT2 and ROBO

Slit2 is a chemorepellent ligand that, by binding to Robo 1 and 2 receptors, participates in axonal growth and branching [193,194,195] and in Schwann cells’ migration [158]. During pancreas development, Slit2 guides the correct migration of pancreatic enteric and sensory nerves [196,197] and has a role in preserving pancreatic progenitor identity [198].

In PDAC tumors, Slit2 expression is reduced [84]. Interestingly, restoring Slit2 expression decreases metastatization and neural invasion by reducing chemoattraction between Schwann cells and cancer cells, thus indicating that acting on Slit2 and Robo 1 signaling might prevent PNI [84].

### 6.4. Cellular Adhesion Molecules

Cellular adhesion molecules have important roles in tumor progression and metastasis in several cancers [199,200] and are also likely implicated in enabling cancer–nerve contact.

#### 6.4.1. Neural Cell Adhesion Molecule 1

Neurons and developing Schwann cells express NCAM1, which has a crucial role in neural growth, adhesion and regeneration [201,202,203]. Indeed, after nerve injury dedifferentiated Schwann cells re–express NCAM to support neuronal growth and axonal guidance [204].

Elevated histological expression of NCAM correlates with increased PNI [150] and decreased patient survival [151]. Further, recent studies suggest that upregulation of NCAM1 in Schwann cells could promote cancer cells’ migration and dispersion, suggesting a role for NCAM1 in directing the migration of neoplastic cells towards nerves [92]. Moreover, upregulation of NCAM1 has been associated with a reduction in E–cadherin mediated cellular adhesion, thus increasing neoplastic cell migration [149].

#### 6.4.2. L1 Cell Adhesion Molecule

L1CAM is a transmembrane neuronal protein important for neuronal migration and differentiation [152,205,206]. It is highly expressed in pancreatic cancer cells and its levels of expression correlate with cancer progression, metastasis, pain and PNI [152,153,154,155,156]. It has been recently proposed that homotipic L1CAM–L1CAM interaction between Schwann cells and PDAC cells could promote nerve invasion. These molecules could indeed attract cancer cells to intrapancreatic nerves, upregulate MMP–2 and MMP–9 expression along the axons to facilitate extracellular matrix breakdown and cancer progression [207].

#### 6.4.3. Mucin 1

MUC1 is a transmembrane protein regulating both adhesive and antiadhesive properties between cells [208] and it is overexpressed in pancreatic cancer [209,210,211]. Elevated levels of MUC1 might provide adhesive advantage to cancer cells favoring their survival inside the nerves by binding to Myelin associated glycoprotein on Schwann cells [88]. Further, augmented MUC1 expression correlates to increased metastasis and poor prognosis in humans [157,158,159,160,161,162]. Interestingly, MUC1 overexpression confers chemoresistance to genotoxic anticancer treatments [212,213].

### 6.5. Matrix Metalloproteinases

Matrix Metalloproteinases (MMPs) are proteolytic enzymes required for the degradation of extracellular matrix; several MMPs are implicated in PDAC [214], in particular high levels of MMP–2/9 in PDAC correlate with metastasis and poor prognosis [215,216,217,218]. MMP–2/9 are also expressed in Schwann Cells, where they facilitate neuritic growth and regeneration after injury [120,219] and cleave beta dystroglycan, altering the structure of the axonal compartments [220].

Interestingly, GDNF upregulates MMP–9 expression [163], while NGF–TrKA signaling promotes MMP–2 activity and expression [164]. All these studies further underscore the crucial role of MMPs and neurotrophin signaling in PNI [165].

### 6.6. Neurotransmitters

Neurotransmitter signaling can modulate different aspects of cancer development, such as proliferation, invasion, metastasis [221,222,223,224]. (See BOX 2 for autonomic pancreatic innervation [225,226]).

Sympathetic signaling via norepinephrine has been shown to increase in vitro neural invasion in DRG neurons by activating the STAT3 β–adrenergic receptor (signal transducer and activator of transcription 3). Further, they upregulate MMP–2/9 in cancer cells, increasing their motility [227,228,229]. On the contrary, GABA—the negative regulator of β–adrenergic signaling—reduces cancer cell proliferation and reverses the pro–growth effects of nicotine in PDAC xenografts models [230,231,232].

In addition, high adrenergic catecholamines levels promote NGF and BDNF secretion favoring tumor innervation, ensuing further adrenergic signaling in a feed forward loop that promotes tumor growth. Blocking this axis acting on β–adrenergic receptor 2 or Trk receptor reduces tumor innervation and prolongs mice survival [233]. Accordingly, recent evidence suggests that usage of β–blockers may increase survival in PDAC patients [234,235,236].

Of note, ablation of sensory neurons in a mouse model of PDAC reduced the initiation and progression of PDAC [55]; nevertheless, the mechanisms by which denervation alters tumor progression is not yet clear, thus requiring additional studies to better comprehend the complexity of PNI.

While sympathetic signaling promotes PDAC, parasympathetic signaling could inhibit PDAC progression. In animal models, vagotomy accelerates pancreatic cancer development, which is in turn inhibited by cholinergic muscarinic agonists. These most likely downregulate the inflammatory stimuli and inhibit TAMs and TNFα release [237]. Notably, muscarinic signaling via the M3 muscarinic receptor CHRM1 suppresses PDAC growth [238].

Clinical findings have also reinforced these results as vagal nerve activity has a protective role in patients affected by metastatic PDAC [239].

PDAC cells also overexpress the NK–1R receptor, to which Substance P—a neuropeptide released by small–diameter sensory C fibers—can bind [136,240]. Recently, it has been proposed that substance P could promote cancer cell proliferation and neural invasion and may also have a role in pain onset [241,242].

Box 2—Pancreatic innervation

In pancreatic cancer there is a decrease in sympathetic fibers, while no difference was noted among parasympathetic fibers [225,226,243].

Sympathetic innervation originates from the sympathetic preganglionic neurons in the lower thoracic segments of the spinal cord; they exit through the sympathetic ganglia without synapse, form splanchnic nerves and terminate on celiac ganglia. Catecholaminergic neurons of these ganglia innervate the intrapancreatic ganglia, islets, blood vessels and the ducts and acini.

Parasympathetic fibers instead originate from the vagus nerve, enter the pancreas through the celiac plexus and end on intrinsic ganglia; they activate parasympathetic post–ganglionic neurons in pancreatic ganglia primarily via signaling on nicotinic acetylcholine receptors.

Sensory information from the pancreas is transmitted to the central nervous system via both vagal and spinal pathways.

### 6.7. Translational perspectives

Given the strong clinical impact, PNI represents an attractive therapeutic target. However, while there are no active clinical trials on PDAC patients, a few pre–clinical studies on animal models have targeted the main molecules involved in PNI (Table 2). For example, it has been shown that NGF inhibition reduces cancer progression [112,133,233], similarly blocking NT–3–TrkC signaling slows PDAC growth in murine models [125]. Moreover, targeting the GDNF–RET axis as well as LIF inhibition reduces PNI both in vitro and in vivo [64,89], while targeting Neurturin impairs PNI and cancer aggressiveness [128]. In addition, it has been recently proposed that radiation might reduce PNI by acting both on neoplastic and nervous cells [244]. Though promising, all these targets have been investigated only in pre–clinical settings and further studies are needed to translate their feasibility in clinical practice.

## 7. Conclusions and Future Directions

PNI is a complex phenomenon that involves multi–directional communication between nerves, cancer and cells of the neoplastic microenvironment. The interactions occurring in PNI reciprocally support nerve and cancer growth, dispersion and migration. Interestingly, the neuroplasticity events of PDAC partly resemble the physiological mechanisms occurring after nerve injury. Thus, a combined effort between scientists and experts in all these fields might help in unraveling the pathological mechanisms underlying PNI and possibly develop new therapeutic approaches for this devastating cancer.

## Figures and Tables

**Figure 1 cancers-11-00893-f001:**
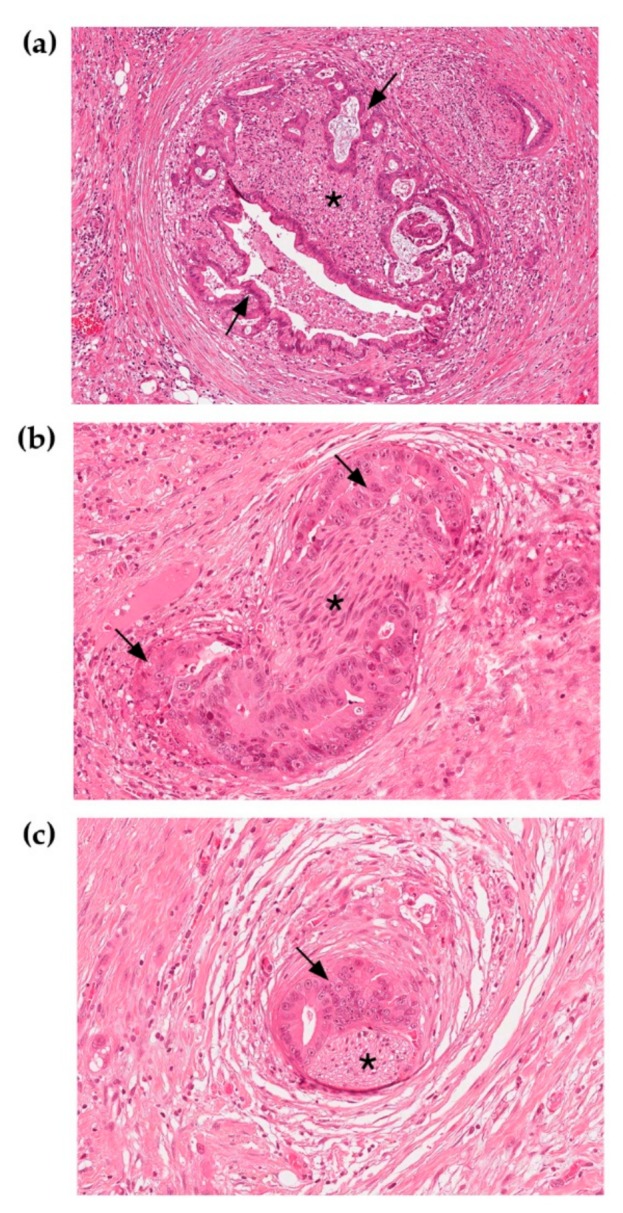
Nerves invaded by pancreatic ductal cancer stained with Hematoxylin and Eosin. (**a**) Neoplastic glands are growing around and within the nerve fiber. (**b**), (**c**) Small diameter nerve fibers surrounded by neoplastic glands. 20× Magnification. Arrows indicate neoplastic glands; asterisks point to nerve fibers.

**Figure 2 cancers-11-00893-f002:**
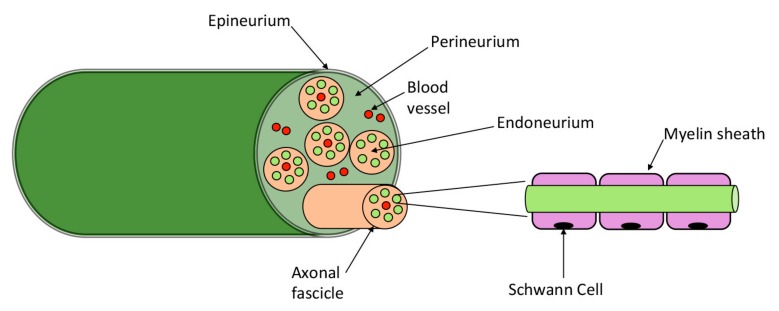
Peripheral nerve structure. The endoneurium is the innermost layer and consists of nerve fibers, composed of axons surrounded by different subtypes of Schwann Cells (myelinating and non–myelinating Schwann cells); it also contains blood vessels, resident macrophages and fibroblasts [56]. The perineurium, which surrounds the endoneurium, is a layer of cylindrical cells tightly interconnected with tight and gap junctions, forming a protective barrier [57]. The epineurium is the outermost layer surrounding several nerve bundles; it includes an elastin and collagen sheath, blood and lymphatic vessels, resident macrophages and mast cells, fibroblasts. The epineurium is itself–innervated by small axons deriving from the endoneurium [58].

**Table 1 cancers-11-00893-t001:** Molecules involved in PNI.

Class of Molecules	Molecule	Role in PNI	Refs
Axonal guidance molecules	SLIT2–ROBO	Increased expression correlate with PNI and metastasis	[80,84]
Interleukin	LIF	Secreted by CAFs, increases neural plasticity and PNI	[89]
Chemokines and receptors	CCL2–CCR2	Recruits TAMs to cancer site promoting PNI via GDNF–RET signaling	[56,115,116,117,118]
	CXCR1–CX3CL1	Increases PNI and promotes cancer–nerve adhesion	[20,121,122,123]
Neurotrophinsand receptors	NGF–TrkA/p75NTR	Increases growth, proliferation and nerve–cancer affinity	[7,18,51,116,124,125,126,127,128,129,130,131,132,133,134,135,136]
	BDNF–TrkB	Increases in vitro proliferation; linked to metastasis	[137,138,139]
	NT3–TrkC	Increases cancer invasiveness, possible role in pain generation	[124,125,140,141]
GDNF family and receptors	GDNF–GFRα1–RET	Increases PNI; TAMs promote PNI through GDNF secretion	[2,64,83,142,143,144]
	Neurturin–GFRα2	Increased expression correlates to enhanced PNI	[128,145]
	Artemin–GFRα3	Promotes cancer migration and PNI	[146,147,148]
Cellular adhesion molecules	NCAM	Reduces cell—cell adhesion; increased expression contributes to PNI and metastasis	[92,149,150,151]
	L1CAM	Mediates homotipic interactions between cancer and nerve increasing PNI; upregulates MMP2–9 expression facilitating cancer progression	[152,153,154,155,156]
	MUC1	Increased levels promote cancer adhesion to nerves and metastasis	[88,157,158,159,160,161,162]
MMPs	MMP2–9	Increases PDAC aggressiveness	[163,164,165]

**Table 2 cancers-11-00893-t002:** Molecules involved in PNI–translational perspectives.

Target	Intervention	Effect	Refs
LIF	Antibodies against LIF	Reduction of neural density and nerve infiltration	[89]
CCR2	Inhibition of CCR2 in combination with FOLFIRINOX	Safe and feasible; not yet investigated specifically for PNI	[118]
NGF	Depletion by anti–NGF antibodies; gene silencing with siRNA	Reduction in progression, neural invasion, metastasis and pain	[112,133,233]
NT3–TrkC	Antibodies against NT3	Slower PDAC growth	[125]
GDNF–GFRα1–RET	GFRα1 silencing by siRNA, antibodies against RET. Radiation	Block endoneural macrophages activation and cancer invasiveness. Decrease GDNF secretion	[64,244]
Neurturin–GFRα2	Depletion by anti–NRTN antibodies	Decrease in neural density	[128]
β2adrenergic receptor (ADRB2)	Pharmacological blockage	Decrease cancer–nerve interactions; diminish neurotrophins secretion	[233]

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
