# Peer review of "Nerves and Pancreatic Cancer: New Insights into A Dangerous Relationship"

_cancers, 2019, doi:10.3390/cancers11070893_

Reviewer 1 Report

This is an interesting and comprehensive review on nerve-tumor cell interactions, elucidating the mechanisms of the high incidence of perineural infiltration by pancreatic ductal adenocarcinoma.

Author Response

“This is an interesting and comprehensive review on nerve-tumor cell interactions, elucidating the mechanisms of the high incidence of perineural infiltration by pancreatic ductal adenocarcinoma”.

We thank the reviewer for his/her comments and for having found our review of broad interest.

Reviewer 2 Report

The manuscript by Gasparini and colleagues reviews the interaction of nerves and pancreatic cancer cells. The topic is timely and of great interest. This expert review is well written and provides an extensive overview of the topic. There are only a few minor topics to address:

1.       A schematic drawing of a nerve would be helpful (endoneurium, perineurium, epineurium etc.).

2.       “PNI is much less frequent in Intraductal Papillary Mucinous Neoplasm (IPMNs)”: does this refer to invasive IPMN, i.e. pancreatic cancer arising from IPMN or IPMN as a precursor? This should be clarified.

3.       A short paragraph about the most promising therapeutic targets of cancer-nerve interaction and planned or active clinical trials could be added.

Author Response

The manuscript by Gasparini and colleagues reviews the interaction of nerves and pancreatic cancer cells. The topic is timely and of great interest. This expert review is well written and provides an extensive overview of the topic.

We thank the Reviewer for this comment. We have addressed the majority of his/her concerns as detailed below.

1. A schematic drawing of a nerve would be helpful (endoneurium, perineurium, epineurium etc.).

We have added a schematic figure of a myelinated peripheral nerve as requested.

2. “PNI is much less frequent in Intraductal Papillary Mucinous Neoplasm (IPMNs)”: does this refer to invasive IPMN, i.e. pancreatic cancer arising from IPMN or IPMN as a precursor? This should be clarified.

We agree with the Reviewer that this sentence is unclear. We thus modified it as follows: “PNI is less frequent in Intraductal Papillary Mucinous Neoplasms (IPMNs) with invasive carcinoma”.

3. A short paragraph about the most promising therapeutic targets of cancer-nerve interaction and planned or active clinical trials could be added.

We have included a brief paragraph about the possible therapeutic targets of PNI. At the moment, we are not aware of clinical therapeutical approaches that have entered or are in clinical trials, as evidenced also in searching Clinical Trail databases. Therefore, we have focused on preclinical models and studies.

Reviewer 3 Report

The authors have written a well-researched and thorough review on perineural invasion (PNI) in PDAC progression and prognosis. I have couple of comments that will improve the review.

Comments:

1)     Schematic – A nice graphic on the different signaling and molecular interactions between the different cell types in the tumor and the pancreatic neural plexis will greatly add to the review. It will make things clear and easy to digest.

2)     Line 92- Add small paragraph on comment on why PNI is present only in PDAC and not IPMNs?

3)     Line 98- I don’t see much controversy. The first few lines in Section 2 has some references that go against PNI in PDAC but rest of the content in this section aligns with prevalence and bad prognosis of PNI in PDAC. So I suggest, either remove that phrase or indicate other contradictory findings.

4)     4.2- Invitro models: Add couple of sentences on which mouse model is better for studying PNI- KPC or KIC. Advantages/disadvantages of each

5)     Add reference (Line 164)- studies that have employed orthotropic models to study PNI

6)     Line 217- What is the consequence of having enhanced neurite growth (in relation to PNI)

7)     Line 279- These references are general for activity and downstream pathways activated by neurotrophins. But what happens specifically in PDAC? Are there papers showing specific effects in PDAC.

8)     Add another table (Table 2) summarizing potential therapeutic points of intervention to prevent PNI. There are references throughout the text (for eg. 418 beta- blockers may increase survival in PDAC patients), but a nicely concise table delineating targets that could be exploited in the clinic would be of interest to the readers.

Author Response

The authors have written a well-researched and thorough review on perineural invasion (PNI) in PDAC progression and prognosis.

We thank the reviewer for his/her comments. What follows is a detailed answer to his/her comments.

1) Schematic – A nice graphic on the different signaling and molecular interactions between the different cell types in the tumor and the pancreatic neural plexis will greatly add to the review. It will make things clear and easy to digest.

We thank the reviewer for the suggestion. We also thought of implementing the review with a scheme depicting all molecules. However, since their cellular localization and temporal expression is extremely diversified and complex, we considered that any kind of picture would have not been both inclusive and complete. Thus, we summarized all of them in Table 1.

2) Line 92- Add small paragraph on comment on why PNI is present only in PDAC and not IPMNs?

In this paragraph, that we rephrased based on rev #2 suggestions’ we comment that PNI is also present in IPMNs, though it is less frequent. We did not further stress this point, as there is no extensive literature on this topic. Furthermore, PNI has been mainly described in degenerated IPMNs that will progress towards cancer.

3) Line 98- I don’t see much controversy. The first few lines in Section 2 has some references that go against PNI in PDAC but rest of the content in this section aligns with prevalence and bad prognosis of PNI in PDAC. So I suggest, either remove that phrase or indicate other contradictory findings.

We rewrote this paragraph to better clarify the concept. We hope the reviewer will find it better specified as detailed in manuscript.

4) 4.2- Invitro models: Add couple of sentences on which mouse model is better for studying PNI- KPC or KIC. Advantages/disadvantages of each

Though we have included specific advantages of these two models in Box-1, we added a couple of sentences in the text as requested.

5) Add reference (Line 164)- studies that have employed orthotropic models to study PNI.

As requested we added references for orthotopic PNI models.

6) Line 217- What is the consequence of having enhanced neurite growth (in relation to PNI)

In this paragraph, we refer to the signaling between macrophage-Slit3 and Schwann cells-Robo, which is the only proven mechanism thus far implicated in physiological conditions to direct nerve elongation. In line with the interactions between immune, glial and tumor cells occurring in PNI, we think it could be worthy to investigate possible alterations in this signaling that might also aberrantly influence nerve spreading in PNI in PDAC.

7) Line 279- These references are general for activity and downstream pathways activated by neurotrophins. But what happens specifically in PDAC? Are there papers showing specific effects in PDAC.

We agree with the reviewer that this is a general part on neurotrophins. To better highlight the role of these molecules in PNI we have detailed their role in the following paragraphs.

8) Add another table (Table 2) summarizing potential therapeutic points of intervention to prevent PNI. There are references throughout the text (for eg. 418 beta- blockers may increase survival in PDAC patients), but a nicely concise table delineating targets that could be exploited in the clinic would be of interest to the readers.

As asked also by reviewer #2 we have included a paragraph on possible therapeutical interventions. Since there are no active clinical trials on patients, we have limited the description to pre-clinical analyses. In addition we also included a new Table, Table 2, as requested by this reviewer summarizing all above described targets.

Reviewer 4 Report

Thank you for the opportunity to review the article. Couple of suggestions for authors

The article is well written, but certain phrases are repeated. Perhaps a professional English editing would help.

The topic of PNI has become extensively researched in recent times. Perhaps adding a line about what this review article would bring to the table which other reviews do not, in other words what could be a unique selling point of this review, would help the reader understand the value of this well researched paper. 

Line 57-58: There argument starts with citing a retrospective study from the same author which did not find correlation between PNI and survival. PNI is a known adverse prognostic marker where presence of PNI is associated with higher local recurrence and shorted DFS. The paragraph between 64-70 is actually contradictory to this statement as well. In my opinion, authors should pitch this paper in a different fashion where they should discuss the latest comments about PNI.

Line 98-100: The opening paragraph argues about utility of using PNI as a marker to decide the kind of treatment for pancreatic cancer. There are many factors which go into deciding treatment for a tumor, especially for an aggressive malignancy like pancreatic cancer. AJCC 8th edition does not acknowledge PNI in the TNM system. I would urge the authors not to present outdated information to pitch an argument. 

In the beginning of the article it is always nice to add a paragraph about how the literature was searched and which databases were used to procure articles for this review. 

Author Response

Thank you for the opportunity to review the article. The article is well written, but certain phrases are repeated. Perhaps a professional English editing would help.

We thank the reviewer for his/her comments. As suggested, we have performed English editing to make the manuscript more readable.

Reviewer #4:

Thank you for the opportunity to review the article. The article is well written, but certain phrases are repeated. Perhaps a professional English editing would help.

We thank the reviewer for his/her comments. As suggested, we have performed English editing to make the manuscript more readable.

The topic of PNI has become extensively researched in recent times. Perhaps adding a line about what this review article would bring to the table which other reviews do not, in other words what could be a unique selling point of this review, would help the reader understand the value of this well researched paper.

We have explained what we consider the novelty of this review at its beginning.

Line 57-58: There argument starts with citing a retrospective study from the same author, which did not find correlation between PNI and survival. PNI is a known adverse prognostic marker where presence of PNI is associated with higher local recurrence and shorted DFS. The paragraph between 64-70 is actually contradictory to this statement as well. In my opinion, authors should pitch this paper in a different fashion where they should discuss the latest comments about PNI.

We agree with the reviewer that this paragraph was unclear. We have modified it accordingly. We hope that the new version will clarify the argument as requested also by reviewer #3.

Line 98-100: The opening paragraph argues about utility of using PNI as a marker to decide the kind of treatment for pancreatic cancer. There are many factors which go into deciding treatment for a tumor, especially for an aggressive malignancy like pancreatic cancer. AJCC 8th edition does not acknowledge PNI in the TNM system. I would urge the authors not to present outdated information to pitch an argument.

We agree with the Reviewer that this sentence was confusing and we have deleted it from the revised version of the manuscript.